# Exploring the perceptions of living donation among potential Moroccan donors in the Netherlands

Charifa Zemouri[1], Assia Nait Kassi[2], Muriel Sinselmeijer[3], Bert Elbertse[4], Yvonne Mulder[3], Sara Jobse[5], Eva-Maria Merz[6,7]*

1 Zemouri et al., Amsterdam, The Netherlands, 2 MentalEdGroup, Abu Dhabi, United Arab Emirates, 3 Department of Research & Laboratory Services, Sanquin Blood Supply Foundation, Amsterdam, The Netherlands, 4 Matchis Foundation, Leiden, The Netherlands, 5 Dutch Transplant Foundation, Leiden, The Netherlands, 6 Donor Behaviour Group, Sanquin Research Amsterdam, Amsterdam, The Netherlands, 7 Department of Sociology, Vrije Universiteit Amsterdam, Amsterdam, The Netherlands

* e.merz@sanquin.nl

## Abstract

### Introduction

The Moroccan-Dutch population is underrepresented in Dutch donor pools, threatening access to matched blood products and transplants. This study explored Moroccan-Dutch individuals willingness to donate blood, stem cell, and live organ donation, and preferred information dissemination methods.

### Methods

A qualitative and exploratory ethnographic study was employed using a survey followed by a focus group to examine willingness and information needs regarding living donation (blood, stem cells, kidney, liver) among Moroccan-Dutch individuals aged 18–55. Survey results were thematically analysed to inform an interview guide for in-depth focus group. Eight participants (four male, four female) were purposefully invited. The session was recorded, transcribed, and analysed thematically.

### Results

Surveys from 119 persons and one focus group were analysed. Ten themes emerged across tissue types: 1) awareness of need; 2) information and knowledge; 3) donation process; 4) cues to action; 5) attitude; 6) religious legitimacy; 7) health challenges; 8) fear; 9) social cohesion and solidarity; 10) relationship with recipient. Participants were willing to donate blood, but needed reminders to prompt registration. Hesitation around stem cell donation stemmed from unfamiliarity and Islamic uncertainty. For living organ donation, emotional ties to recipients influenced willingness. Most participants were unaware of ethnic matching needs and shortages, and

**Data availability statement:** Participants consented to data collection and analysis for publication purposes, but were not asked for permission to share their raw data. Furthermore, the data are qualitative and in Dutch, with a strong cultural and religious context that may lead to misinterpretation if removed from its original setting. However, de-identified data can be made available upon motivated request, subject to approval. The raw data are owned and stored by Matchis and the first author (contact@matchis.nl; charifazemouri@hotmail.com).

**Funding:** This work was funded by the Dutch Ministry of Health, Welfare, and Sport (Project Grants), case number 90001539. The funders had no role in study design, data collection and analysis, decision to publish, or preparation of the manuscript.

**Competing interests:** The authors have declared that no competing interests exist.

preferred campaigns highlighting these aspects. A multi-component communication strategy was preferred, combining medical, technical, religious, and emotional information, including patient stories, to address gaps and insecurities.

## Conclusion

Participants were unfamiliar with stem cell donation, living organ donation, and ethnic matching. They were willing to donate tissue if aware of the shortage, the need for ethnic matching, and religious legitimacy They sought medical-technical information from donor organisations but preferred religious legitimacy communicated by an Islamic scholar. Participants stressed the need for a tailored campaign addressing barriers, ethnic matching, and reliance on the same community for successful living donation.

## Introduction

Living human donations are possible for blood products, stem cells and specific organs, impacting survival rates and treatment of several diseases such as cancer or kidney failure. However, to have a successful match between donor and recipient, matching based on antigen types is necessary [1]. This diversity can be reached by increasing the number of potential donors from different ethnic backgrounds. Most large donor banks are situated in Europe and North America and include mostly white Western European donors while having a high ethnically diverse patient population within their countries [2]. Therefore, they cannot always meet the clinical demands of their residing ethnic groups [3,4]. One such country is the Netherlands, with approximately 450,000 blood donors, 415,000 registered stem cell donors and 533 annual living organ donors in a population of 18 million in 2024, of which approximately 26% has a non-Dutch ethnical background [5–8].

The exact estimates of donor diversity are unknown because the collection institutions in The Netherlands do not register ethnicity. Based on ethnicity data from scientific research, it is estimated that Dutch citizens (with citizenship or who reside in the Netherlands) of Moroccan, Turkish and Suriname descent are twice as likely to not register as organ donors and are less registered in live kidney donation [7,9]. In the Dutch stem cell registry, it was estimated that 95% of registered stem cell donors are of North-West European origin [5]. Furthermore, the Dutch National Blood Supply reports that blood donors with an ethnic minority background are underrepresented whilst a higher diversity is needed to meet blood demands [10]. This scarcity is the main challenge for meeting the transfusion and transplantation demands of patients with a diverse background where ethnic matching is crucial for decreasing adverse reactions and increasing survival [2,11,12].

One of the largest ethnic minorities in the Netherlands is of Moroccan descent with approximately 420.000 citizens with and without Dutch citizenship (from here on referred to as Moroccan-Dutch). A considerable proportion of this group will eventually need human tissue transfusion or transplantation if any physical complications

occur [9]. The Moroccan population has a high ethnic diversity which poses a challenge in finding suitable donors [13]. The chance of finding a suitable sibling donor is only 50%, leading to reliance on national donor registries for non-sibling donors. When it comes to stem cells, Moroccan-Dutch individuals rely on the Dutch stem cell registry for stem cell transfusion, as Morocco does not have a national registry [14]. The exact number of Moroccan-Dutch individuals who are living donors is unknown and assumed to be low and insufficient for their healthcare needs.

Moroccan-Dutch patients who need tissue or organs must rely on other Moroccan-Dutch individuals willingness to donate blood and stem cells or consider living organ donation. Without, Moroccan-Dutch patients may face health consequences and long waiting times for treatment. The shortage of donors may result in ethnic health disparities and poor health outcomes for this group. Therefore, the Dutch Ministry of Health requested the Dutch National Blood Supply (Sanquin), the Dutch Centre for Stem Cell Donors (Matchis), and the Dutch Transplantation Society (Nederlandse Transplantatie Stichting, NTS) to study the perceptions and information needs of the Moroccan-Dutch population to inform and develop evidence-based campaigns. A qualitative study was conducted with the aim to explore Moroccan-Dutch individuals views on living blood, stem cell and living organ donation, their willingness to donate and their preferred methods of information dissemination.

## Methods

### Study design

This study employed a qualitative and exploratory ethnographic design to comprehensively analyse participants' opinions and insights on living donation by first rolling out an online questionnaire followed by a focus group [15]. The manuscript was completed and assessed with the consolidated criteria for reporting qualitative research (COREQ-32) checklist for qualitative research [16].

The study setup and execution were preceded and enhanced with public and patient involvement (PPI) by forming an advisory group of three Moroccan-Dutch professionals active in their community [17]. C.Z. selected purposefully the advisory group members based on their active community participation and professional expertise. The group (N = 3) included a 35-year-old female behavioural scientist, a 39-year-old male imam and Islamic theologist, and a 24-year-old male law student and columnist.

The advisory group supported the development of culturally appropriate and comprehensible online questionnaires and distributed them within their networks. They also helped to identify missing codes and interpret themes from the questionnaire data. The group helped recruit the participants for the focus group. The focus group topic list with in-depth questions was co-created with the advisory group. The advisory group was financially compensated for their contributions.

### Study setting, population and recruitment

This study was conducted in preparation of a nationwide targeted donor recruitment campaign of Sanquin, Matchis, and NTS. The study took place in the Netherlands, and targeted Dutch individuals who self-identify as Moroccan with or without a citizenship, inviting them to participate by completing a questionnaire or join a focus group. The eligibility criteria for participation were ages 18 to 55 (eligible age for stem cell donation), identifying as Moroccan, and being proficient in Dutch to complete the Dutch online questionnaire. The focus group participants were purposefully recruited to ensure diversity in sex, age and educational background, and diversity in attitudes (positive, negative, ambivalent) towards the topic by asking the persons in advance about their stance. Persons with a medical profession were excluded from participation to prevent bias and steering of opinion in group dynamics. The focus group was held in Dutch in a community cultural centre in Amsterdam, New West.

### The online questionnaire

The study gathered participants' views through an online questionnaire developed using Qualtrics. Initially, the questionnaire was piloted within the researchers' network and the advisory group to refine its content and format. The

questionnaire began with a cover page explaining the study's purpose, identifying the funding institutes (Sanquin, Matchis, NTS), and defining "living donation" to avoid any confusion with post-mortem donation.

Participants were required to select a consent button before proceeding, ensuring informed agreement to participate. The questionnaire included closed-ended questions to collect demographic data and open-ended questions for participants to share their experiences and opinions in detail (see Supplementary File 1 S1 The online questionnaire). The question framing regarding donation status was based on the registration system and regulations in the Netherlands. Questions about blood and organ donations were in retrospect. Those who are registered as blood donors have typically already donated at least once. In contrast, the Netherlands does not have a registery for living organ donation, so we asked participants about this form of donation in retrospect. For stem cell donation, the focus is on registration because individuals can register as potential donors, but only a small number will ever be matched and asked to donate. Our interest lied in those who are registered for the diversity of the donor database, rather than those who have already donated.

The questionnaire link and accompanying information were shared via WhatsApp and direct messaging on social media platforms like X, Instagram, and LinkedIn to reach the target population. Potential participants were encouraged to forward the questionnaire to their networks to broaden their reach. Invitations to participate in focus groups were distributed through similar channels.

## Focus group

One focus group was conducted with eight participants (four men and four women) to gain in-depth insights, perspectives, and opinions from the participants, complementing the questionnaire. The online questionnaire results informed the focus group guide, probing questions and topics for the focus groups in collaboration with the advisory group. The session began with general introductions and discussions on the willingness to donate blood, stem cells, and organs during life. C.Z. facilitated the discussions in Dutch and took field notes. C.Z. had no relationships with any of the participants. Y.M., B.E., and S.J. supported C.Z. for additional in-depth questions. The focus group session took two hours and data saturation was achieved when all the topics were discussed, the focus group fully elaborated on their views and no more questions for clarification arose. The discussion was recorded using Philips voice tracer DVT2050, transferred to an MP3 file and secured for data safety. The recordings were transcribed using Amber Script and manually corrected by C.Z. for analysis, after which the recordings were deleted. The focus group participants received financial compensation for their collaboration and did not receive the transcripts for comments.

## Data analysis

The questionnaires remained online until data saturation was reached, defined as the point where no new themes or insights emerged responses [18]. Incomplete questionnaires, such as those without answers to open-ended questions or containing responses like 'none' or 'not applicable,' were excluded from the analysis. Closed questions were analysed using descriptive statistics, while free-text responses and focus group input were subjected to thematic analysis following the six-step plan of Kiger and Varpio [19]. This process involved familiarising with the data, coding key segments, identifying and refining themes, and defining and naming themes. The questionnaire responses and focus group transcripts were anonymised and analysed inductively, with themes organised into broader categories addressing blood, stem cells, and organs. Participant quotes were selected to illustrate key themes and translated into English for reporting. The translations were done by C.Z. and validated by A.N.K. who is also an English language academic editor and further verified by other co-authors. Thematic analysis was not conducted on information needs due to limited generated data, as responses were brief, repetitive, or lacked variation. Instead, the data was summarised narratively to ensure accurate representation and inclusion in the analysis.

The analysis was conducted by C.Z. and reviewed by A.N.K., with any discrepancies resolved through consensus and consultation with the advisory group. The advisory group also ensured completeness by reviewing for missing codes or themes and verifying cultural accuracy in the interpretation of findings.

### Researcher reflectivity and positionality

C.Z. and A.N.K., both women of Moroccan descent who were born and grown up in the Netherlands' multicultural environment, led the analysis and interpretation of the data in this study. They are both native Dutch and native Moroccan-Arabic speakers and fully proficient in English. C.Z. and A.N.K. have both strong cultural competence and awareness in relation to the study population because they're part of the Moroccan-Dutch population. Their research roles, combined with their extensive experience and training in qualitative research, particularly with ethnic minority groups, brought valuable insights. However, their professional backgrounds and personal experiences were also considered potential influences on their perspectives. C.Z., a public health researcher, conducted the focus group discussions with support from Y.M., B.E., and S.J., affiliated with a donor registry or collection institution. A.N.K., a behavioural researcher, contributed significantly to data interpretation. C.Z. had professional ties with the advisory group.

### Statement of ethics

**Study approval statement.** The ethical waiver was obtained from non-WMO (Medical Research Involving Human Subjects Act, Netherlands) Committee of the Medical Ethics Review Committee (METC) Amsterdam University Medical Centers under METC number 2023.0949. The waiver was provided on January 4, 2024, after which data collection commenced. The ethical waiver was obtained in written format in Dutch and English and communicated by prof. dr. J.A.M. van der Post, the chair of the committee.

The ethics board waived the study, which was commissioned by the Dutch Ministry of Health, Welfare and Sports. In the Netherlands, the Moroccan community does not have a formal community leader structure. Therefore, no community-level consent was applicable; however, PPI was ensured through an advisory panel, and all participants, both from the PPI group and study sample, provided individual informed consent before participation.

**Consent to participate statement.** All participants of the online questionnaire had to tick the consent box before proceeding. The participants of the focus group were informed in writing about the study and had to sign a written informed consent in person prior to the study.

## Results

### Questionnaire output

We received 125 completed questionnaires within 18 hours, with data saturation being achieved by the 80th response, meaning no new themes, patterns, or insights emerged after that point. We chose to include all 125 responses since this was the total number collected at the time, and we did not want to discard any potentially valuable data. This ensured that the dataset was as comprehensive as possible, despite saturation being reached earlier. Six questionnaires were excluded: not of Moroccan descent, >55 years old, and answering 'none' to all open-ended questions. Thus, 119 questionnaires underwent thematic analysis. Table 1 presents the participant characteristics, and Table 2 provides details about the focus groups.

### Themes

Ten key themes emerged from the thematic analysis of the written responses. Given the interconnected nature and interpretation of the data, some responses could be classified under multiple themes, resulting in potential

**Table 1. Questionnaire response overview and characteristics.**

| Characteristics | | |
|---|---|---|
| Sex | Female | 75 (63.0%) |
| | Male | 44 (37.0%) |
| Age in years | Mean | 35.73 |
| | Standard deviation | 8.48 |
| | Range | 18-55 |
| Ethnicity (self-identified) | Moroccan | 115 (96.6%) |
| | Partially Moroccan (either parent is Moroccan) | 4 (3.4%) |
| Have you ever donated blood? | Yes | 26 (21.8%) |
| | No | 91 (76.5%) |
| | I do not know | 2 (1.7%) |
| Are you a registered stem cell donor? | Yes | 14 (11.8%) |
| | No | 94 (79.0%) |
| | I do not know | 11 (9.2%) |
| Have you ever donated an organ or part of an organ? | Yes | 0 (0%) |
| | No | 119 (100%) |
| **Questionnaire response rate** | | |
| Which reasons do you have for (not) donating blood? | | 100% |
| Which reasons do you have for (not) donating stem cells? | | 100% |
| Which reasons do you have for (not) donating organs? | | 100% |
| Which information do you need about blood donation? | | 66.6% |
| Which information do you need about stem cell donation? | | 70% |
| Which information do you need about living organ donation? | | 66.6% |
| How would you like to be informed? | | 100% |

**Table 2. Focus group participant characteristics.**

| Participant code* | Sex | Age | Educational level | Work setting | Donor status |
|---|---|---|---|---|---|
| V1-S | Woman | 30 | Vocational tertiary education | Library | Considers blood donation and stem cell registration. |
| V2-S | Woman | 39 | Vocational tertiary education | Administration | Donated blood in the past. |
| V3-M | Woman | 19 | Bachelor's | Student | Active blood donor; registered stem cell donor. |
| V4-F | Woman | 50 | Vocational tertiary education | Community work | Not a donor. |
| M1-Y | Man | 37 | Bachelor's | Municipality | Not a donor. |
| M2-M | Man | 45 | Master's | Banking | Not a donor. |
| M3-N | Man | 31 | Vocational tertiary education | Arts and culture | Not a donor. |
| M4-A | Man | 19 | Vocational education | Student | Not a donor. |

\* **Participant coding:** V = female (*vrouw* in Dutch); M = male (*man* in Dutch); the number indicates the participant's order within their gender group, and the final letter represents their first initial. This coding system was chosen based on how participants addressed each other during the focus groups, to allow for clear distinction between individuals in both discussion and transcript.

overlap. However, our thematic analysis intentionally separates these categories to capture how these barriers manifest and influence different aspects of donation behaviour. By maintaining these distinct categories, we provide a more comprehensive understanding of how various barriers interact with specific stages of the donation decision-making process. Table 3 presents these themes, their conceptualisation, and their occurrence across different tissue types.

**Table 3. Overview of overarching themes per types of donation.**

| # | Theme | Conceptualization | Blood | Stem cells | Organ |
|---|-------|-------------------|-------|-----------|-------|
| 1 | Awareness of need | Recognition of the necessity for living donation due to shortages and its importance for patient care. | Yes | Yes | No |
| 2 | Information and knowledge | • **Top down**: information and knowledge shared from the donation organizations.<br>• **Bottom up**: knowledge gaps from the person. | Yes | Yes | Yes |
| 3 | Donation process | Obstacles concerning the donation process split into three subthemes:<br>• **Registration** (e.g. registration, data handling).<br>• **Medical process** (e.g. medical check-up, tissue extraction).<br>• **Emotional process** (e.g. emotions linked to the donation process). | Yes | Yes | Yes |
| 4 | Cues to action | Stimulus that can prompt action: donation, self-study, reflection. | Yes | Yes | Yes |
| 5 | Attitude | The attitude of the person towards donation, this may be negative, neutral, ambivalent, or positive. Trust and distrusts are also aspects included as part of an attitude. | Yes | Yes | Yes |
| 6 | Religious legitimacy | Reasons for donation driven from a religious (Islamic) perspective or motivation.<br>• **Barrier**: When a person perceives donation as unlawful by Islamic legislation.<br>• **Facilitator**: When a person perceives donation as a good deed by Islamic legislation. | Yes | Yes | Yes |
| 7 | Health challenges | Health issues that are prevalent or may be perceived as challenge and may prevent a person from donation. | Yes | No | No |
| 8 | Fear | Experienced fear from the person concerning to donation. | Yes | Yes | Yes |
| 9 | Social cohesion and solidarity | Donating can be divided into two main subthemes:<br>• **Altruism**: Helping others without expecting anything in return.<br>• **Reciprocity**: Assisting others with the expectation of receiving help in return or giving back after having been helped.<br>These captures motivations rooted in broader community values, such as a sense of duty or altruism toward the collective. | Yes | Yes | No |
| 10 | Relationship with recipient | Relationship the participant has with the potential recipient. This could be friendship, kinship, romantic or non-existing.<br>In contrast to the previous theme, this theme focuses on the individual relationship a participant may have with a recipient. | No | Yes | Yes |

## Awareness of needs

This theme highlights the participants' identified recognition and awareness of the importance of living donations. Several respondents acknowledged the need for living donors. Some were familiar with the need for sufficient blood and stem cell donors, specifically for the Moroccan population, and the need for ethnic matching. The need for sufficient organs was not mentioned in the questionnaire output. Except for one participant (V2-S), the focus group participants were unfamiliar with shortages of living donors, ethnicity-based shortages, and the need for a diverse donor pool. Only two focus group participants (V2-S, V3-M) knew of blood shortages, having experienced blood donation within the family or learned from Sanquin. Those unaware believed that the collection institutes needed to communicate the shortages actively and emphasise the importance of ethnic matching to increase donation willingness.

*"The difference here is that it is rarer to find a suitable match, and ethnic background plays a significant role. I was once told that in the Netherlands, there is less than a 1 in 1000 chance that someone with a Moroccan background will find a match if they need a stem cell donation." Questionnaire participant, man, 25 years old.*

## Information and knowledge

This theme focuses on information and knowledge gaps related to living donation. The questionnaire revealed an overall knowledge gap on living donation which was related to donation hesitance. The responses showed that multiple respondents were unaware of the possibility of stem cell donation. Regarding living organ donation, lack of knowledge about

living organ donations and potential lifelong consequences contributed to their hesitancy. The focus group mentioned that while blood donation is well-known, shortages and the need for matching are not. Participants' willingness depended on knowledge availability, with many questions about stem cells, registration, donation, and the importance. More awareness of shortages, matching needs, procedures, and risks was mentioned as necessary. The focus group elaborated on the need for more information on matching criteria and post-donation health impacts before considering living organ donation.

*"Start with what stem cells are. (...) There is a significant lack of knowledge. If we start thinking now about how to register, that's step 15, jumping ahead. Step 1 is understanding what a stem cell is." Focus group participant, M1-Y.*

### Donation process

This theme focuses on the organisational, practical and procedural aspects of the donation process. The questionnaire indicated that participants needed familiarity with the donation process, particularly regarding stem cell and living organ donation. They described the registration and potential donation process as administratively burdensome, citing insufficient time and opportunities to understand the procedures. Perceived administrative hurdles and registration costs for stem cells beyond a certain age were mentioned as barriers. Concerns included donor eligibility, potential physical impacts, and recipient outcomes, particularly with bone marrow punctures in stem cell donation. Participants in the focus group also expressed unfamiliarity with living donation procedures, comparing them with post-mortem protocols. They raised inquiries about donation prerequisites, registration procedures, and sources of reliable information as a necessity to consider their donorship. Additionally, they sought clarification on tissue and organ handling, implications of shortages, and impacts on the donors. For living organ donation, questions centred around prioritisation on waiting lists, lifelong dialysis, and logistical aspects of registration and how potential donors are connected to patients in need. The focus group emphasised that perceived barriers to donor registration must be addressed before considering living donation.

*"Although it seems like an exciting procedure to me. From the top of my head, isn't this [stem cell donation] done with a spinal tap? If so, no way Jose." Questionnaire participant, man, 32 years old.*

### Cues to action

Questionnaire responses revealed that a lack of consideration, thought, or discussion was the reason for not considering donation. Many participants had not pondered donation due to a lack of exposure and awareness and desired more cues to action. The focus group emphasised the necessity for increased community outreach on living donation. Participants highlighted that raising awareness stimulates looking up information, contemplation, and decision-making. They stressed the importance of highlighting donor shortages to underscore the urgency of transplantation needs. Motivational factors were identified as catalysts for action, such as community discussions, educational readings on living donation, and considering registration. Similarly, the Dutch national campaign following the post-mortem Dutch Donor Law prompted widespread community discussions and decisions about donor status. The group suggested that similar campaigns are needed to encourage community dialogues about blood and stem cell donations.

*"Until someone in your surroundings becomes ill with leukaemia or sickle cell disease or something similar, and there is suddenly a demand for it [blood]. Then people start to look into it, and you suddenly realize how few people there are from your background, your ethnic background, with stem cells, for example. And then, yes, it suddenly becomes real." Focus group participant V1-S.*

### Attitude

The questionnaire output attitude towards living donation varied per donation type, with an objection-free positive attitude towards blood. The attitude towards stem cell and living organ donation could be described as more hesitant due to unfamiliarity and an unwillingness to donate to strangers. For living organ donation, the responses showed that participants were more willing to donate a liver piece due to its regenerative nature than a kidney. The attitude responses from the

questionnaire were similar to those of the focus group. The participants differed in their attitudes towards stem cell donation, which strongly depended on their available knowledge of the subject. Familiarity with stem cell donation led to a more positive attitude and willingness to consider registration, while unfamiliarity was a reason for hesitation. Some participants noticed that parental attitudes towards stem cell donation and registration also influenced participants' views due to the ethical questions involved in this subject. In general, the attitude towards living organ donation was negative unless the recipient was a relative.

*"I had an Islamic upbringing, and good deeds were always stimulated, but donation was always a bit of a taboo. We never discussed it at home until it became a relevant topic [because of the Dutch Donor Law]." Focus group participant M2-M.*

### Religious legitimacy

Religion, in this case Islam, could be both a facilitating and hindering factor for living donation, which was visible in both the questionnaire and focus group responses. Some questionnaire participants questioned the permissibility of living donation under Islam and needed information about Islamic jurisprudence. Others were religiously motivated and mentioned their willingness to donate 'for God's' sake. Some cited religious concerns about the sanctity of the body in the questionnaire as a potential barrier. Islamic teachings influenced these individuals' willingness to donate by guiding their decisions; if Islam prohibited an action, they would refrain from it, whereas if it permitted the action, it removed a potential barrier to donation. The focus group participants linked their willingness to be living donors to Islamic permissibility. Five out of eight affirmed the acceptance of blood donation in Islam. They acknowledged varying views on the Islamic permissibility of living donation. The Islamic stance on stem cell donation was unknown within the group, but participants saw no religious barrier if permissible. Living organ donation's religious considerations were irrelevant to the group. They mentioned that in this case the relationship with the recipient is important (see theme Relationship with recipient). Participant V2-S doubted the notion that saving a life through living organ donation could be impermissible. These responses reflect the diversity of interpretations within Islamic jurisprudence and highlight the participants' reflections on the religious and ethical implications of donation.

*"Because Allah says in the Quran, if you save one life, it is as if you saved humanity." Focus group participant V1-S.*

### Health challenges

The questionnaire output provided an extensive list of prevalent health problems such as anaemia, fainting, overweight, diabetes, low blood values, hypotension, hypertension, chronic diseases, and medication as perceived reasons for not being eligible as a blood donor. When health challenges were mentioned for stem cell donation, it was from participants who had tried to become a donor but did not meet the medical eligibility criteria. During the focus group, two participants (V2-S and M3-N) shared that experiencing pain and having iron deficiency were their health-related barriers to becoming a blood donor. No other prevalent health barriers to stem cell or living organ donation were mentioned, especially since the participants were unfamiliar with the conditions.

*"I wanted to become a donor. But because I have vitamin D and iron deficiency, I was not eligible." Questionnaire participant, woman, 34 years old.*

### Fear

For all living donations, the answers to the questionnaire revealed fear of needles, surgery, blood, anxiety disorders or panic attacks. Fear significantly hindered the willingness to register for stem cell donation, which was related to possible bone marrow punctures. For living organ donation, fear was related to challenges after donation and implications of living with one kidney. There was no fear for blood donation as a topic in itself, but rather for the necessary use of needles. The focus group shared the same fear of pain and elaborated per tissue type. One focus group participant (V3-M) feared

needles but overcame it by weighing the pros and cons for blood donation. Others did not fear blood donation. The focus group participants were reluctant about bone marrow puncture but more willing to donate stem cells through peripheral blood as it was considered less fear-inducing. The willingness to donate stem cells depended on the recipient of the cells, mentioning that their desire to overcome their fears and pain was greater for immediate family. For living organ donation, fear of health implications, consequences of donation and failure in transplantation were mentioned as barriers.

*"If it hurts, I don't want to do it. But if it can be done through blood, then I will." Questionnaire participant, man, 33 years old.*

### Social cohesion and solidarity

According to the questionnaire and the focus group, the 2023 earthquake in Morocco served as a powerful motivator for (disaster-driven) blood donations. Participants noted that seeing the King of Morocco and other public figures donating blood after the earthquake demonstrated social cohesion and solidarity within their community. This public display of support highlighted a sense of collective responsibility, reinforcing a shared cultural identity and sense of duty to help one another in times of need and served as a trigger for a communal call to action. Additionally, reciprocity, the idea of giving in anticipation of a potential future need for blood or stem cells for oneself, also encouraged donation. Many participants felt that their contributions to the donor pool today could lead to future assistance for themselves or their families, underscoring a form of "mutual aid." The willingness to donate an organ during life was closely linked to the donor's relationship with the recipient, with strong preferences to donate to close family members.

The focus group added that Islamic teachings endorse blood donation as a good deed. Religion here not only motivates individual actions but also binds the community, reinforcing solidarity through the view that donation aligns with the principles of charity in Islam. Regarding living organ donation, in line with the responses from the questionnaire, participants expressed a reluctance to donate an organ to strangers. This hesitation stemmed from concerns about personal health risks and undergoing surgery for unknown individuals. Furthermore, there was an underlying fear of donating to 'the wrong person'. A wrong person was described as a criminal of any sort or someone who does not share the same values or beliefs. It led to the question whether or not that person 'deserves' their organs.

*"Because there are too few donors! And I think it's [stem cells] important that this helps children, but also adults!" Questionnaire participant, woman, 36 years old.*

### Relationship with the recipient

The questionnaire data showed that the emotional connection with the recipient strongly influenced the willingness to donate stem cells or organs during life. In contrast, there was hesitation in donating to unfamiliar individuals due to concerns about compatibility and potential outcomes. The focus group discussion echoed the importance of the relationship with the potential receiver of the organ or stem cells. Participants emphasised a willingness to donate to a mother or child, prioritising familial bonds over religious considerations. Emotional closeness with the recipient was a significant factor. Participants showed reluctance to donate organs to strangers due to personal risk concerns, but they accepted these risks when considering their mothers as recipients.

*"For my mother, I would accept the pain and consequences. But I would have to reconsider for my neighbour or uncle with whom I have no emotional connection." Focus group participant, M2-M.*

### Campaigns and communication

The participants emphasized that providing clear communication and information about the importance of ethnic matching in tissues donation can create unity within the community, as it highlights a shared responsibility to meet specific health needs. They wanted information on shortages, matching procedures, and the lifesaving impact of donations. Participants

sought insights into the consequences of shortages, pros and cons for donors, dispelling potential fears arising from ignorance, and access to expert advice or online resources.

Clear information on Islamic regulations regarding living donation, indicating its religious legitimacy, was a collective information need. Active endorsements from Islamic authorities were deemed crucial. According to the participants, entities like mosque unions should widely endorse messages representing various theologians, mosques, and groups and disseminate them through their networks. Participants suggested that this endorsement should be integral to organisational campaign efforts.

Trust in the entities communicating about donations was considered vital. The participants emphasized the importance of a 'triangular communication approach,' which they described as involving three key groups: medical institutions for providing accurate health information, Islamic theologians for religious guidance, and patients in need of donations to share their personal experiences and emotional perspectives. By incorporating these diverse voices, participants believed this approach would foster greater trust and confidence in the donation process. The participants stressed the influential role of key figures and role models, known for their credibility and community engagement. The group acknowledged the effectiveness of key figures in the community, such as well-known imams or Islamic organizations, famous persons within the community who are perceived as reliable information dissemination and raising awareness, in raising awareness about important issues and believed their involvement would enhance trust in donation information. Additionally, there was a suggestion to find a versatile spokesperson, such as an expert from the community who can function as a bridge between all relevant stakeholders to inform the community about living donation effectively.

Participants expressed a need for information through online channels like social media, Google, websites, or informative emails. There was a demand for active communication, including written materials, general media, and educational efforts. Acknowledging the labour intensity of education on an individual or neighbourhood level, participants suggested communication at large gatherings, such as Friday or Eid prayers, for a wider reach. The 'train-the-trainer' concept was emphasised to ease the burden on organisations. The participants added that communication should be in simple and understandable Dutch, void of academic jargon, and resonating with the target audience. For older audiences, Moroccan-Arabic or Tamazight could be used orally. Word-of-mouth was mentioned as a powerful and effective communication tool.

## Discussion

This qualitative study explored the willingness of Moroccan-Dutch individuals to become living donors of blood, stem cells or organs. The willingness was influenced by awareness, knowledge, fear, health, donation registration, cues to action, religious legitimacy and the relationship with the potential recipient. Participants had positive attitudes towards blood donation but needed to be made aware of the under-representation of ethnically diverse donors. Hesitancy towards stem cell donation stemmed from unfamiliarity and concerns about religious permissibility. Similar to previous studies, willingness to donate organs during live was mentioned to depend on a solid emotional connection with the recipient [20,21]. For all tissues, participants stressed the need for targeted campaigns to communicate the impact of ethnicity on successful matching, preferring information from donation organisations, Islamic theologians or Imams, or emotionally resonant sources. A previous Dutch study among persons with Ghanaian and Surinamese-African backgrounds also indicated the need for an emotional trigger in decision-making towards tissue donation [4].

This is the first study to investigate facilitators and barriers towards living donations across donation types simultaneously. Yet, the results align with previous studies on either type in persons with and without migration backgrounds, but also on Muslim populations for deceased donation [22,23]. Despite their willingness, most participants are currently not active as blood donors due to the lack of awareness of need, a finding consistent with a European Union-wide study and a global review [23,24]. This emphasises the need to create more awareness [24,25]. Participants showed more hesitancy towards stem cell donation due to a need for knowledge, religious legitimacy, and family opinions. A similar study

on factors influencing decision-making in Saudi persons also reported hesitance based on family perceptions [26]. Family opinions and intergenerational familial agreement are significant and play an important role within the Moroccan-Dutch communities and should therefore be considered by stakeholders involved in the donation processes and their governance. These stakeholders, including healthcare providers, community leaders, and religious advisors, should engage with families through culturally sensitive outreach [27,28].

Most Moroccans identify as Muslims, and therefore, religious legitimacy was a recurrent theme throughout the data for decision-making and communication needs. Unlike many studies that identify religion as a negative factor, our data showed Islam as a facilitator for increasing donation willingness [29–31]. When religion was perceived as a barrier, it was due to a lack of knowledge about Islamic legitimacy, as previously reported [32]. Religion was less decisive for organ donors to relatives, a phenomenon noted in several studies among Muslim populations [26,33]. While Islamic scholars and councils, such as the British Board of Imams and Scholars, agree that stem cell donation is permissible, this information is not widely known within the targeted population [23,34,35], highlighting a need for better communication and hence increasing trust in information from collection institutes and generally in the healthcare systems [13,36]. A supporting example was visible in Syria, where kidney transplant rates rose from 7 to 17 per million in 5 years after Islamic religious authorities endorsed post-living and deceased kidney transplants [37,38]. The latter highlights the direct measured impact of communicating religious legitimacy.

Our findings identified a specific communication need within the study population, which has a significant implication for future targeted donation recruitment campaigns in the Netherlands. The campaigns should be tailor-made, specifically designed and customized, to fit the target audience's unique needs, preferences, and cultural context to facilitate an active outreach. Active community outreach was previously proven effective in increasing donor rates in Morocco [36]. The campaigns should consider varying levels of willingness to donate blood, stem cells or living organs. For blood donation, stem cell donation, and living organ donation, raising awareness of shortages based on ethnic matching is necessary. Addressing questions on religious legitimacy and promoting donations as an Islamic good deed, supported by Islamic entities and real-life stories, may meet information needs. For example, in the Islamic Republic of Iran, religiously based communication regarding blood donation resulted in an emotional bond with the blood bank and a willingness to donate [39]. A systematic review of facilitators and barriers for minority group blood donations also revealed the importance of religion, culture and family opinion, especially when the eligible populations need to identify themselves with the subject [22]. Furthermore, the study population required emotional reactions and reasoning to their will to donate. Emotional triggers and reactions have been studied before and are a significant factor for decision making [31,40]. Finally, receiving information from medical professionals with similar ethnic backgrounds could be of additional value [4]. Therefore, a multifaceted communication approach considering emotional, rational, and religious components may be required to recruit more donors of Moroccan descent. The effect of the to be developed and implemented campaign should be studied by monitoring the effects and responses from the community.

### Strength and limitation

The main strength of this study is its sequential qualitative data collection method. The online questionnaire gathered data from a large sample, providing a comprehensive overview of opinions on living donation. Key issues and patterns identified in the questionnaire were explored in-depth during the focus group, resulting in a thorough understanding of the study population's knowledge, views and needs. Our PPI advisory board contributed to sound interpretation of data. Previous studies have shown that involving PPI improves the usefulness of the data and fits the population's perspective [41,42]. Additionally, this study provided a comprehensive overview of barriers, facilitators, and information needed to tailor recruitment efforts for more living blood, stem cell and organ donors.

The primary limitation was a potential sampling bias for the questionnaire participants and focus group. The high percentage of respondents who reported being blood or stem cell donors exceeds the national average, indicating a possible

bias [5–8]. Furthermore, persons not interested or with negative attitudes towards donation likely did not participate, leading to missing negative attitude responses. The first two authors' Moroccan-Islamic background provided valuable insights for data interpretation but could introduce bias by approaching the data based on readily available knowledge. To counteract this bias, the ethnically diverse research team critically appraised the output. The questionnaire and focus groups were conducted in Dutch, so opinions from those not proficient in the language might have been missed. Therefore, the discussion leader actively asked about the perceptions of the participants' families and circles to obtain views on social norms regarding living donations [43].

## Conclusion

The Netherlands has a shortage of Moroccan living donors, challenging the medical needs of Moroccan-Dutch patients due to ethnic matching. Increasing the donor pool is crucial to reduce ethnic health disparities. Therefore, this study explored the motivation, willingness, and preferred information needs of living donation. Participants were willing to donate tissue during their lives if they were aware of the shortage, the need for ethnic matching, and the religious legitimacy of donation. They sought information from donor organisations on medical-technical aspects. Future campaigns should consider a multifaceted approach, including the need for ethnic matching for successful tissue transplantation, emotional appeals to trigger an emotional reaction and religious legitimacy to boost registration and commitment.

## Supporting information

**S1. The online questionnaire. The original questions in Dutch are reported with the translations to English.**
(DOCX)

**S2. Inclusivity-in-global-research-questionnaire_CZ.**
(DOCX)

## Acknowledgments

The authors would like to express their gratitude to the advisory group for their invaluable PPI contributions and insights throughout the study, as well as to the participants of the questionnaire and focus groups. Additionally, we extend our gratitude to New Metropolis West in Amsterdam for providing the venue for the focus group session. We also like to thank Jaap Dijkman from Matchis Foundation for his support for setting up this project.

## Author contributions

**Conceptualization:** Charifa Zemouri, Bert Elbertse, Yvonne Mulder, Sara Jobse, Eva-Maria Merz.

**Data curation:** Charifa Zemouri.

**Formal analysis:** Charifa Zemouri, Assia Nait Kassi.

**Funding acquisition:** Bert Elbertse, Yvonne Mulder, Sara Jobse.

**Investigation:** Charifa Zemouri.

**Methodology:** Charifa Zemouri, Eva-Maria Merz.

**Project administration:** Muriel Sinselmeijer, Bert Elbertse, Yvonne Mulder, Sara Jobse.

**Resources:** Charifa Zemouri, Muriel Sinselmeijer, Bert Elbertse, Yvonne Mulder, Sara Jobse.

**Software:** Charifa Zemouri.

**Supervision:** Muriel Sinselmeijer.

**Validation:** Assia Nait Kassi, Eva-Maria Merz.

**Visualization:** Charifa Zemouri, Bert Elbertse, Yvonne Mulder, Sara Jobse.

**Writing – original draft:** Charifa Zemouri.

**Writing – review & editing:** Assia Nait Kassi, Muriel Sinselmeijer, Bert Elbertse, Yvonne Mulder, Sara Jobse, Eva-Maria Merz.

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
