## [Decision Letter · Decision Letter 0]

Dear Dr. Merz,

Thank you for submitting your manuscript to PLOS ONE. Before it is accepted for publication, we invite you to submit a revised version of the manuscript that addresses the suggestions for improvement from the two reviewers.

We look forward to receiving your revised manuscript.

Kind regards,

Alberto Molina Pérez, Ph.D.

Academic Editor

PLOS ONE

Journal Requirements:

This work was funded by the Dutch Ministry of Health, Welfare, and Sport (Project Grants), case number 90001539.

7. Please remove all personal information, ensure that the data shared are in accordance with participant consent, and re-upload a fully anonymized data set.

Reviewers' comments:

Reviewer's Responses to Questions

**Comments to the Author**

1. Is the manuscript technically sound, and do the data support the conclusions?

Reviewer #1: Yes

Reviewer #2: Yes

2. Has the statistical analysis been performed appropriately and rigorously?

Reviewer #1: N/A

Reviewer #2: Yes

3. Have the authors made all data underlying the findings in their manuscript fully available?

Reviewer #1: Yes

Reviewer #2: Yes

4. Is the manuscript presented in an intelligible fashion and written in standard English?

Reviewer #1: Yes

Reviewer #2: Yes

Reviewer #1: Reviewer Comments

Dr Britzer Paul Vincent

Firstly, I would like to thank the editor and authors for the opportunity to review this manuscript titled “Exploring the perceptions of living donation among potential Moroccan donors in the Netherlands.” This is a highly valuable and timely piece of work, and I am pleased to note that it was commissioned by the Dutch Ministry of Health, Welfare, and Sport. The methodological rigour demonstrated reflects the considerable effort invested in this study. The findings related to religious legitimacy particularly resonate with the findings of our recently published systematic review, “Barriers and facilitators of deceased organ donation among Muslims living globally: An integrative systematic review.”

I offer the following minor but important comments for consideration prior to acceptance for publication:

1. Terminology Consistency: Research involving ethnically diverse populations is vital. However, for the research to be both accurate and meaningful, the terminology used to describe these populations must be consistent and clearly defined. In this manuscript, the study population is referred to variously as ‘Moroccan donors in the Netherlands’ and ‘Moroccan-Dutch population’. Using different terms to refer to the same population may cause confusion. It is recommended that one term is used consistently throughout the manuscript. If these terms are intended to denote different groups, please provide a clear rationale and definition. For instance:

o Moroccan-Dutch could refer to individuals of Moroccan origin who hold Dutch citizenship or reside in the Netherlands and identify as having both Dutch and Moroccan cultural affiliations.

o Moroccans living in the Netherlands might encompass a broader term encompassing all Moroccans in the Netherlands, including those who may not be Dutch citizens or who do not identify as Moroccan-Dutch.

This point is particularly important, as inconsistencies in ethnic classification have been shown to present limitations in systematic reviews and scoping studies (see: [Review 1: Barriers towards deceased organ donation among Indians living globally]; [Review 2: Barriers to conversations about deceased organ donation among adults living in the UK]).

2. Line 75–76: The statement “A considerable proportion of this group will eventually need human tissue transfusion or transplantation” should be supported by a relevant reference. If available, please provide an estimate of the prevalence of end-stage organ failure among this population to substantiate the claim.

3. Line 122: Please clarify how “Dutch individuals of Moroccan descent” were identified. Were participants asked to provide any documentation to confirm their ethnic background or nationality? If not, kindly state that participants self-identified as Moroccan-Dutch/Dutch individuals of Moroccan descent/Moroccans living in the Netherlands (the term confusion aligns with the earlier point regarding the need for consistent and well-defined terminology.)

4. Line 127: About recruitment, when mentioning “diversity in attitudes toward the topic”, please explain how this variation was assessed or identified during participant selection.

5. Line 129–130: Please specify the language(s) in which the survey and focus group discussions were conducted. Also, indicate who facilitated the focus groups and comment on their fluency or proficiency in the relevant language(s).

6. Line 151–152: Please provide a rationale for conducting mixed-gender focus groups. Given the sensitivity of the topic and its cultural and religious dimensions, it is common practice to conduct gender-segregated groups to promote openness and to mitigate power or speech dynamics that may inhibit discussion.

7. Line 161–162: Again, kindly specify the language used for both the survey and focus groups. If translation was required, please describe how it was conducted and verified.

8. Line 178–179: This line suggests that English was not the language used for data collection. It is essential to state earlier in the methods section the language used, the facilitator’s language proficiency, and the process followed for translation and verification.

9. Reflexivity Section: It would strengthen the methodological transparency of the study to include information regarding the researchers’ language proficiency and their cultural competence or awareness in relation to the study population.

10. Line 224–225: Please revise the sentence for clarity: “Tables 1 and 2 present the characteristics of survey respondents and focus group participants, respectively.”

11. Table 1: Please define what is meant by “partial Moroccan”. Does this refer to mixed heritage or other criteria?

12. Table 1: The question on stem cell donation appears to ask about registration, whereas the questions on blood and organ donation refer to actual donation history. Please explain this discrepancy or justify this variation in framing.

13. Table 1: The response to the question “Have you ever donated an organ or part of an organ?” shows n = 100 for No, yet the percentage is listed as 0%. Kindly correct this inconsistency.

14. Table 2 – Education Level: Please clarify the meaning of “University of applied sciences”. Do you mean vocational tertiary education? Also, consider aligning educational categories with international equivalents such as Bachelor’s, Master’s, and PhD degrees for clarity.

15. Line 239: The term “frequency of occurrence” may imply quantitative enumeration. Consider rephrasing to simply state “…and their occurrence across different tissue types.”

16. Recommendations Table: As stated in line 92, one of the fund’s aims was to “inform and develop evidence-based campaigns.” In light of this, it would be beneficial to conclude the manuscript with a table of actionable recommendations, derived from the ten identified themes. This would be of practical value for policymakers and campaign developers working with this population in the Netherlands.

Reviewer #2: This is a well written clearly structured and highly relevant manuscript that effectively explores the perceptions of living donation among potential Moroccan donors in the Netherlands. The message is portrayed clearly and this study offers original and valuable insights pertinent to the clime. The article is relevant to this specific population group studied.

I commend the authors for their adherence to the COREQ guidelines, which significantly enhances the transparency and rigor. The methodology section is detailed and explicit, clearly providing a road map of the research process. The data presentation is clear and accessible making the findings easy to interpret.

Comments for Consideration

1. Participant codes in Table 2 - please can you clarify if these codes depict any specific meaning specifically what do the letters after the numbers signify? (e.g., " V1-S", " V3-M") it is OK if there is no significance.

2. Table 3 caption- write out the caption for types of donation within the table as this will ensure that anyone viewing the table as a standalone can easily understand the categories of donations presented without needing to refer back to the main text.

**Do you want your identity to be public for this peer review?** For information about this choice, including consent withdrawal, please see our Privacy Policy

Reviewer #1: **Yes: ** Britzer Paul Vincent

Reviewer #2: No

---

## [Author Response · Author response to Decision Letter 1]

21 Jun 2025

We addressed all editor and reviewer comments in a separate file, attached as 'Response to reviewers'. As we give answers to every comment in a table format, we cannot paste the text here, as it would ruin its readibility.

---

## [Editor Report · Decision Letter 1]

Exploring the perceptions of living donation among potential Moroccan donors in

the Netherlands

PONE-D-25-23161R1

Dear Dr. Merz,

We’re pleased to inform you that your manuscript has been judged scientifically suitable for publication and will be formally accepted for publication once it meets all outstanding technical requirements.

Kind regards,

Alberto Molina Pérez, Ph.D.

Academic Editor

PLOS ONE

Additional Editor Comments (optional):

Congratulations for this well designed and well conducted study! Also, I wish to thank the two reviewers for their thourough review and constructive comments.